# Plasma Surface Modification of 3Y-TZP at Low and Atmospheric Pressures with Different Treatment Times

**DOI:** 10.3390/ijms24087663

**Published:** 2023-04-21

**Authors:** Sung Un Kang, Chul-Ho Kim, Sanghyun You, Da-Young Lee, Yu-Kwon Kim, Seung-Joo Kim, Chang-Koo Kim, Hee-Kyung Kim

**Affiliations:** 1Department of Otolaryngology, School of Medicine, Ajou University, Suwon 16499, Republic of Korea; 2Department of Molecular Science and Technology, Ajou University, Suwon 16499, Republic of Korea; 3Department of Chemical Engineering, Department of Energy Systems Research, Ajou University, Suwon 16499, Republic of Korea; 4Department of Chemistry, Department of Energy Systems Research, Ajou University, Suwon 16499, Republic of Korea; 5Department of Prosthodontics, Institute of Oral Health Science, School of Medicine, Ajou University, Suwon 16499, Republic of Korea

**Keywords:** plasma gases, zirconium oxide, surface properties, atmospheric pressure, vacuum

## Abstract

The efficiency of plasma surface modifications depends on the operating conditions. This study investigated the effect of chamber pressure and plasma exposure time on the surface properties of 3Y-TZP with N_2_/Ar gas. Plate-shaped zirconia specimens were randomly divided into two categories: vacuum plasma and atmospheric plasma. Each group was subdivided into five subgroups according to the treatment time: 1, 5, 10, 15, and 20 min. Following the plasma treatments, we characterized the surface properties, including wettability, chemical composition, crystal structure, surface morphology, and zeta potential. These were analyzed through various techniques, such as contact angle measurement, XPS, XRD, SEM, FIB, CLSM, and electrokinetic measurements. The atmospheric plasma treatments increased zirconia’s electron donation ($\gamma^{-}$) capacity, while the vacuum plasma treatments decreased $\gamma^{-}$ parameter with increasing times. The highest concentration of the basic hydroxyl OH(b) groups was identified after a 5 min exposure to atmospheric plasmas. With longer exposure times, the vacuum plasmas induce electrical damage. Both plasma systems increased the zeta potential of 3Y-TZP, showing positive values in a vacuum. In the atmosphere, the zeta potential rapidly increased after 1 min. Atmospheric plasma treatments would be beneficial for the adsorption of oxygen and nitrogen from ambient air and the generation of various active species on the zirconia surface.

## 1. Introduction

Plasma is a state of matter that consists of a quasi-neutral gas of positively and negatively charged particles, including electrons and ions that exhibits collective behavior. It can be artificially generated by applying electric and/or magnetic fields to a neutral gas [1]. Plasma may consist of radicals, electrons, ions, photons, and ultraviolet radiation [2]. In plasma, collisions between various species can mediate the transfer of energy and momentum depending on their temperature and density [3]. At high density, an inelastic collision with the exchange of potential and kinetic energies between plasma components occurs through the excitation and ionization of orbital electrons [3]. Controlling those collision processes can play an important role in plasma applications.

Plasma surface modification techniques have been widely used to functionalize various biomaterials by enhancing their wettability, biocompatibility, or bonding efficiency without changing the bulk properties of the materials [4]. Surface modifications can be obtained through the interaction between reactive species and material surfaces. The mechanisms of plasma include a chemical effect and a physical effect on the surface. The energetic ion bombardment of plasma can modify the surface texture of the target materials, while chemically active free radicals and byproducts play an important role in plasma chemistry [5]. Therefore, a plasma surface modification process can be employed to tailor the surface properties by selectively changing the surface chemistry and morphology [6]. 

Plasma systems can be classified into thermal plasmas and non-thermal plasmas depending on the temperature of the gas. While different types of plasma have their unique applications and advantages, non-thermal plasma, which does not have thermal equilibrium among electrons, ions, and neutral particles, has attracted great interest, particularly in the modification of biomaterial surfaces. This is due to its ability to avoid thermal damage to the surfaces [7]. Non-thermal plasmas can be ignited either at low pressure or atmospheric pressure. In a low-pressure plasma system, gas is excited in a vacuum chamber by supplying electrical energy. This produces energetic ions, electrons, and other reactive particles in a controlled manner [8]. In addition, vacuum ultraviolet emissions in a low-pressure system can contribute to energy transport in the plasma discharge [9]. Compared to vacuum systems, atmospheric-pressure plasmas may offer some technological and economic advantages. These advantages include lower production costs, higher electron collision frequencies, and no need for complicated vacuum equipment [1]. Recently, atmospheric-pressure plasmas have been widely applied in the biomedicine, textile, and food industries [10]. In an atmospheric plasma system, the collisions of gas particles with ions or electrons are frequent due to their short mean free path [11], while it is easier for plasma molecules to diffuse further in a vacuum system [12].

Among structural ceramics, 3 mol% yttria-stabilized tetragonal zirconia polycrystal (3Y-TZP) has been widely used as a dental biomaterial. Its superior mechanical strength, corrosion resistance, high biocompatibility, and good optical properties make it a suitable choice for the manufacture of dental crowns, bridges, inlays, implants, and orthodontic brackets [13]. 3Y-TZP does not trigger any adverse systemic reactions due to its chemically inert nature, but its high resistance to acid etching would limit a chemical bond with resin cements [14]. Hence, it can be a great challenge to obtain strong and durable adhesive bonding of resin cements to zirconia restorations. Zirconia implants have recently gained attention due to the increasing demand for more aesthetic and hypoallergenic treatments. Concerning osseointegration and implant materials, zirconia has been proposed as a good alternative to conventional titanium implants [15]. Surface topography and chemistry play pivotal roles in bonding performance and cell–material interactions. Therefore, particular attention was paid to improving the surface polarity and hydrophilicity of zirconia due to its inherent bio-inertness.

Recently, the surface functionalization of zirconia using plasma technologies has been introduced to alter its surface properties for specific applications [15,16,17]. Herein, plasma processing would create chemically active species on the zirconia surface, which would impart new surface characteristics to enhance its final performance. The nitrogen functionalities on the zirconia surface introduced by plasma-immersion ion implantation (PIII) could increase the antibacterial potential and osseointegration behavior of zirconia [18,19]. Our previous study revealed that the largest increase in the surface energy of 3Y-TZP was obtained with nitrogen/argon mixture plasma generated at atmospheric pressure among several different gases (Ar, N_2_, He/O_2_, and N_2_/Ar) [17]. In N_2_/Ar mixture plasma, a small addition of N_2_ in Ar gas plasma would enhance the production of reactive species or radicals, which can play a significant role in surface chemistry. A previous study demonstrated that the electron density (*n_e_*) increased while the electron temperature (*T_e_*) decreased as the concentration of N_2_ increased from 0 to 10% in N_2_/Ar plasma at vacuum pressure [20]. 

A considerable advantage of the plasma surface modification of 3Y-TZP is that the surface bioactivity can be promoted selectively while the bulk properties of the material remain unchanged. In particular, the successful implementation of plasma technology highly depends on the optimization of the process variables, such as the discharge power, chamber pressure, process gas, treatment time, and gas flow rate [21]. However, there are currently no protocols for plasma surface treatments designed to improve the biological compatibility of 3Y-TZP. In this study, we especially focused on the effect of chamber pressure and plasma treatment time on the surface functionalization of 3Y-TZP with N_2_/Ar gas mixtures. The surface modification of 3Y-TZP was performed using a DBD plasma system at atmospheric pressure and an inductively coupled plasma (ICP) system at low pressure with different treatment times. The null hypothesis was that there would be no significant differences in the surface properties of 3Y-TZP between the different treatment methods (pressure and treatment time).

## 2. Results 

### 2.1. Surface Free Energy (SFE) Components 

A two-way analysis of variance (ANOVA) revealed that there was a statistically significant interaction between chamber pressure and plasma treatment time in relation to the surface contact angle (*p* < 0.05). The changes in the water contact angle as a function of the plasma treatment time are shown in Figure 1. The contact angle decreased with increasing treatment time for both plasma systems, but the atmospheric plasma groups exhibited lower values in the region of 44.8–64.7° than those for the vacuum groups. The surface energy components of all experimental groups based on the probe liquids are shown in Table 1. When calculated using the three-liquid method, the SFE of 3Y-TZP increased with plasma treatments in all groups, with the contribution of an increase in the polar component ($\gamma^{AB}$). The total surface energy is a consequence of the electrodynamic interaction, dominated by acid–base interactions rather than dispersion forces. There was a slight increasing trend in the SFE with increasing plasma treatment time. In particular, the highest SFE value was obtained in V15. It was observed that the plasma treatment on the zirconia surface increased its electron donation ($\gamma^{-}$) capacity under atmospheric pressure conditions, while the plasma treatment decreased its $\gamma^{-}$ parameter in a vacuum system with increasing treatment time.

### 2.2. Surface Chemistry

Figure 2A shows the X-ray photoelectron spectroscopy (XPS) C1s spectra of all experimental groups. The spectrum of the control specimen can be decomposed into three components: a component at 284.8 eV due to the C-C bond from the adventitious carbon layer, a component at 286.4 eV due to the C-O bond, and a component at 288.4 eV due to the O=C-O bond. As the plasma treatment time increased, the intensity of the C-C peak decreased while the intensity of C-O and O=C-O increased, indicating that the zirconia surface was oxidized by the reactive oxygen species (ROS) after irradiation with the atmospheric plasma. The plasma ions broke the C-C bond to form C radicals, which combined with ROS such as O^+^, O^2+^, O_2_^+^, and O_2_^2+^ [23]. As shown in Figure 2C,E,F, atmospheric plasma induced the generation of higher concentrations of oxygen functional groups, which could involve C-C bond breaking compared to vacuum plasma [24].

The N 1s photoelectron region (Figure 2B) revealed a characteristic component at a binding energy of 400 eV, which is assigned to N in zirconium oxynitride or ZrO_x_N_y_. This suggests that zirconium oxynitride was formed in the near-surface regions of all plasma groups. The V20 group showed a bonding configuration of N in zirconium nitride (ZrN) at a binding energy of 396 eV, indicating the formation of a ZrN layer. As previously noted [25], the N concentration in V20 may have reached the critical concentration necessary for ZrN layer formation. 

The XPS spectrum of O 1s is shown in Figure 2C. The peak at ~530 eV belongs to the lattice oxygen (O_L_) in the ZrO_2_, while the peaks at ~531.5 and ~532.5 eV are attributed to the O component associated with acidic hydroxyl OH(a) and basic hydroxyl OH(b) groups, respectively. Figure 2F shows the quantitative ratio of lattice oxygen (O_L_), OH(a), and OH(b) in the O 1s core level spectra for all experimental groups. The atmospheric plasma promoted the formation of the OH(b) group, which is the chemisorbed surface oxygen on the zirconia surface (Figure 2F). The binding energy of O_L_ shifted to a lower energy with the use of vacuum plasma (Figure 2C), implying that the lattice oxygen became more ionic in character [26].

The chemical composition (of a region about 1 μm in depth) obtained from energy dispersive X-ray spectroscopy (EDS) analysis showed the presence of O, C, Zr, and Y elements on the surfaces (Figure 2D). According to the scanning electron microscopy (SEM)–EDS results, the amount of O element increased while the amount of C element decreased in the atmospheric plasma groups. The SEM–EDS analysis limits nitrogen detection due to the low efficiency of low-Z elements [27]. In the XPS results (Figure 2E,F), an increase in the percentage content of the O element with increasing treatment time was observed in the atmospheric plasma groups. The XPS, which is a surface-sensitive technique with an estimated penetration depth of a few nanometers [28], indicated that the N element was implanted on the outermost surface layer following plasma treatments. The changes in N atomic percentages as a function of plasma treatment time are presented in Figure 2G. The N concentrations rapidly increased after 1 min and subsequently decreased in both plasma systems. For the vacuum system, the N atomic percentage decreased to a value of 1.51% and then increased. The in-depth distribution of elements within the near-surface zone of the specimens is given in Appendix A. It can be seen that the N species are incorporated down to ~17 nm, depending on the plasma type and plasma exposure time. The vacuum plasma treatment and increased treatment time tend to enhance nitrogen diffusion.

### 2.3. Surface Characterization 

Figure 3 demonstrates the surface texture parameters (Sa, the arithmetical mean height; Sq, the root mean square height; and Sv, the maximum pit height) of all test groups obtained through the use of confocal laser scanning microscopy (CLSM). All of the plasma-treated groups showed a slight decrease in values, and thus, it could be considered a mild etching effect to smooth the surface, although it is very low. In addition, no statistically significant differences in the Sa and Sq values between the plasma-treated groups were observed, even if the vacuum plasma groups exhibited a small reduction in roughness values.

SEM micrographs (Figure 4) showed that the degradation of the grain boundaries under high electric fields occurred in the vacuum plasma groups. The SEM images also revealed surface erosion due to energetic ion bombardment from high-power pulsed plasma streams. However, as shown in cross-sectional focused ion beam (FIB) images, those changes were confined to the outermost surfaces (≈top 10 nm), and the plasmas did not cause any subsurface damage in all experimental groups. In contrast to vacuum plasmas, the grain boundaries were clearly seen, and some small particles on the grain boundaries were observed at longer exposure times with atmospheric plasmas.

### 2.4. Phase Transformation

The X-ray diffraction (XRD) data were analyzed by the Rietveld method. As shown in Appendix A, the Rietveld refinement results showed that all experimental groups were composed of four different crystalline phases: tetragonal’, tetragonal, cubic, and monoclinic phases. The crystal structures of each phase are shown in Appendix A. To investigate the phase evolution affected by the plasma treatment conditions, the XRD patterns of all experimental groups in the 2θ range of 27° to 31° are depicted in Figure 5A. Asymmetrical broadening and a decrease in the intensity of the tetragonal peak (011)_t_ can be observed in the A1 and V1 groups, suggesting a superposition of two peaks (the tetragonal peak (011)_t_ at 2θ = 30.17° and the cubic peak (011)_c_ at 2θ = 30.07° (JCPDS card 27-0997) [29]. The quantitative phase composition as a function of the exposure time, deduced from the Rietveld analysis, is shown in Figure 5B,C. The control group mainly consisted of tetragonal and cubic phases. In the Rietveld refinements, the cubic phase fraction significantly increased after 1 min of plasma exposure and was further reduced to a control level with increasing exposure times. Figure 5D shows the change in the unit cell parameter (Å) of the cubic phase as a function of the plasma exposure time, and the highest increase in unit cell volume was observed in the A1 group.

### 2.5. Change in Zeta Potential

A two-way ANOVA revealed that there was a statistically significant interaction between chamber pressure and plasma treatment time for the surface zeta potential (*p* < 0.05). The changes in zeta potential as a function of the treatment time with vacuum and atmospheric plasma systems are shown in Figure 6. The untreated specimen (control) showed a negative zeta potential (−28.44 mV), and the plasma treatment increased the zeta potential of 3Y-TZP. In the vacuum plasma groups (V5, V10, V15, and V20), the specimens exhibited positive zeta potential values, which may be attributed to an increase in basic hydroxyl (OH)_b_ groups on the surface. With atmospheric plasmas, the value of the zeta potential became less negative as a function of the treatment time. The zeta potential suddenly increased after 1 min of treatment and continued to increase slightly with longer exposure times. 

## 3. Discussion

This study evaluated the effect of chamber pressure and plasma exposure time on the surface characteristics of 3Y-TZP with N_2_/Ar mixtures as the feed gas for plasma generation. Recently, the plasma nitriding of 3Y-TZP has been performed to produce zirconium nitride (ZrN) in an attempt to enhance its mechanical and optical properties [30]. Pilz et al. [31] identified that 2.5 µm of ZrN coating on the cobalt–chromium–molybdenum orthopedic implant enhanced its antibacterial effects. Milani et al. [25] reported that an approximately 500 µm ZrN layer on the 3Y-TZP surface was obtained with a plasma nitridation time of 120 min at 1450 K in air. Our XPS experimental results confirmed that a ZrN layer formed on the surface of the zirconia after a 20 min vacuum plasma treatment, while zirconium oxynitride formed in other groups where the N concentration was not high enough to convert it into the ZrN structure [25]. During the plasma process, the energetic electrons could excite nitrogen molecules to separate the strong N≡N bond, and thus, the released nitrogen could further react with oxygen atoms [32]. In this study, high-energy plasma in the vacuum chamber improved the NO_x_ production, although the N atomic percentage decreased after 5–10 min and subsequently increased in a vacuum. The decreasing and then increasing tendency found in the vacuum plasma treatment might depend on the chamber volume and the pumping speed. According to the result of this study, the nitridation of zirconia by plasma irradiation is only a surface effect with a limited penetration depth of a few nm, indicating that it would be difficult to introduce nitrogen into a zirconia crystal lattice. However, the vacuum plasma system and increased plasma exposure time could increase the depth of nitrogen diffusion in 3Y-TZP. Yu et al. [33] reported that the incorporation of nitrogen atoms in the TiO_2_ lattice could be enhanced by thermal treatments. Therefore, the effect of thermal treatments on the plasma nitriding process of 3Y-TZP needs to be further investigated.

For surface charges, we measured surface electrokinetic potentials (zeta potentials) based on electrophoretic mobility [34]. The most negative value (−28.44 mV) was obtained on the non-treated surface (control). Both vacuum and atmospheric pressure plasmas increased the zeta potential of 3Y-TZP. The zeta potentials became positive after the vacuum plasma treatments, while the zeta potentials became less negative after the atmospheric plasma treatments. With vacuum plasma, the zeta potential continued to increase until 10 min, after which it decreased. Our results agree with the previous study undertaken by Neelakandan et al. [32], which demonstrated that conductivity is only marginally affected by nitrogen concentrations between 2 and 4% and that the higher concentrations of nitrogen decreased electron mobility. In this study, it was observed that the zeta potential rapidly increased after 1 min of exposure to atmospheric plasmas. Afterward, a slight increase was detected over time. Miyake et al. [35] also found an increase in the zeta potentials of 3Y-TZP after atmospheric plasma treatments. This suggests that basic hydroxyl OH(b) groups would play an important role in changing the zeta potentials through physicochemical modifications. Feng et al. [36] reported that basic hydroxyl groups and the polar components of the biomaterial’s surface could greatly influence cell–material interactions, and thus, positively charged surfaces may enhance the adsorption of serum proteins. 

The XPS results of this study revealed that the atmospheric plasma treatment increased basic hydroxyl OH(b) groups, with the highest percentage of OH(b) group in A5. The results relating to the surface energies showed that both vacuum and atmospheric pressure plasmas increased the polar component ($\gamma^{AB}$), with the greatest increase in V15 followed by A10. Furthermore, the atmospheric plasma treatments increased the electron-donation ($\gamma^{-}$) capacity of 3Y-TZP, while the vacuum plasma treatments decreased $\gamma^{-}$. Increased oxygen adsorption with increased plasma exposure times might contribute to increased electron-donation capacity because oxygen is more electronegative than nitrogen.

In this study, we used N_2_/Ar gas mixtures with a 10% concentration of N_2_ as a plasma source. In the N_2_/Ar non-thermal plasma system, the dominant process to control electron density (*n_e_*) and electron temperature (*T_e_*) would be electron impact collisions and energy transfer through Pennig ionization. Based on our results, those diverse collision processes and interactions can be enhanced with increased plasma exposure time. However, after a certain degree of excitation is acquired, frequent collisions could lead to the slowing down of the hot electrons by the recombination and charge neutralization of the ions [37]. Furthermore, excessive exposure could induce increased trap density, causing a reduction in breakdown strength. In this study, we used a DBD plasma system operated at a low frequency under atmospheric pressure and an ICP system operated at a radio frequency under low pressure. At low pressure, ion bombardment [38] and UV photons [9], can both play essential roles in plasma–material interactions, although this study did not measure UV photons. The electron temperature (*T_e_*) can be enhanced under vacuum conditions due to its strong dependence on the electron mean free path on kinetic energy. In previous investigations related to plasma sterilization [9,39], they demonstrated that continuous pumping in the vacuum chamber could remove by-products without re-contamination. The high-energy electron beam in a high vacuum system can be suitable for ion beam etching or doping processes [40]. Based on the results of this study, the vacuum plasma treatment can be effectively used as a surface nitriding technique through high-energy ion bombardment. However, SEM/FIB images showed surface damage caused by energetic particles, although confined to the outermost surface layer. Therefore, when a vacuum plasma system is used, the ion bombardment energy flux should be controlled to prevent surface damage. 

In this study, as opposed to the vacuum plasma system, a DBD reactor operated at atmospheric pressure produced a higher percentage of reactive oxygen species (ROS) due to collision mechanisms in the air [41]. Sardella et al. [42] reported that the generation of both reactive oxygen species (ROS; H_2_O_2_) and reactive nitrogen species (RNS; NO_2_^−^) by atmospheric plasmas could improve cellular behavior. The study also suggested that both ROS and RNS had a synergistic effect in triggering cell functions. In this study, an increase in oxygen contents, especially OH radicals, at atmospheric pressure might be due to the adsorption of oxygen from ambient air. However, in terms of energy efficiency for gas conversion, the DBD plasma was considered low since the changes in zeta potential were limited, maintaining negative values. Instead of a DBD system, a microwave or a gliding arc discharge system could provide much better energy efficiency at atmospheric pressure due to high vibrational kinetics [41]. Furthermore, to enhance the plasma's performance, the material properties should be considered. Ouyang et al. [43] suggested that the thicker substrates increased the penetration depth of reactive oxygen and nitrogen species (RONS) with atmospheric plasma.

Tuning material properties using plasma treatments depends on the generation of highly reactive plasma species. The reactive species can be produced when the impinging energy of the electron reaches a certain threshold for ionization [44]. Hence, various plasma operating conditions (e.g., gas composition, pressure, power, and flow rate) contribute to characterizing the ion flux–energy distribution function [45]. In this study, we chose 150 W, −200 V_bias_ AC, at 4 Pa as the operating conditions of the vacuum plasma based on previous studies dealing with plasma-enhanced atomic layer deposition using RF-ICP sources. In their studies, moderate plasma power in the range of 100–300 W was used during plasma exposure [45,46]. In this study, the highest O content, the lowest C content, and NH_3_ production were detected in V1. There might be an energy transfer by the high-energy ion beam for the bond breakage (dissociation process) after 1 min of plasma treatment. With longer exposure times, plasma-induced physical or electrical damages, although confined to the outermost layer (≈top 10 nm), were observed due to high-energy ions. Therefore, we presume that a short exposure time (<1 min) would be beneficial for low-damage surface treatments and, at the same time, for the saturation of electromagnetic ions. Alternatively, Park et al. [47] suggested that a two-step N_2_/Ar plasma process could reduce the surface damage caused by Ar ion bombardment. Jung et al. [48] reported that low ICP power (50 W) could accelerate high-energy electrons due to the thick sheath. 

In this study, after 1 min of plasma exposure in air, the lattice oxygen in the zirconia crystal decreased while the O component associated with the O^2−^ ions in surface oxygen vacancies (XPS peaks at ~531.5 eV) increased. The generation of oxygen vacancies, created by the adsorption of CO_2_ [49], changed the electrical properties of 3Y-TZP, which was confirmed by the results relating to the zeta potentials. In comparison with the atmospheric plasma, the formation of oxygen vacancies was significantly promoted after 1 min plasma exposure at low pressure, not so much due to intensive collisions between electrons but rather due to the energetic ion bombardment. 

The substitution, hydration, and redox processes in the zirconia crystal structures could all change the chemical compositions, leading to chemical lattice strain or chemical expansion [49]. In the XRD results of this study, the cubic phases increased after 1 min of plasma exposure in both vacuum and atmospheric plasma systems, which might be responsible for the enhanced oxygen vacancy formation. As reported in Kalite et al.’s study [50], the presence of oxygen vacancies would contribute to stabilizing and generating cubic phases by suppressing the soft X_2_^−^ vibration mode, which corresponds to the cubic-to-tetragonal phase transition. Furthermore, strain-induced tetragonal peak broadening was observed in the A1 and V1 groups. This might be caused by oxygen uptake and/or nitrogen ion implantation in the zirconia structures [51,52]. With longer exposure times, lattice reconfiguration and relaxation might occur due to nitrogen doping in the crystal lattice [53]. Therefore, it can be estimated that the ionization process was highly accelerated after 1 min of plasma exposure in both plasma systems.

Taking all the results into account, both vacuum and atmospheric plasma treatments changed the surface energy, the chemical composition, and the zeta potential of 3Y-TZP. In a vacuum plasma system, the energetic ion bombardment of highly reactive species may play an important role in altering the surface properties of zirconia. In a vacuum plasma system, a much higher conversion and/or energy efficiency could be reached in only a short exposure time (<1 min). However, with a longer exposure, undesirable surface melting or degradation would be induced. In most vacuum plasmas, the collision frequency would be smaller than that of atmospheric plasmas [54]. In an atmospheric plasma system, the plasma–material interaction may be attributed to the generation of reactive oxygen and nitrogen species (RONS) due to electron collisions, a reaction with the surrounding air, and the energy transfer to the zirconia surface, although the energy of atmospheric plasmas would be relatively weaker than that of vacuum plasmas [55]. Atmospheric plasma treatments would be beneficial in terms of the adsorption of oxygen and nitrogen from ambient air and the generation of various active species on the zirconia surface, leading to the production of more active species than under a vacuum. The observed increase in ROS with increased treatment times in this study was in line with the results of previous studies. Gjika et al. [56] found that 30–90 s plasma treatments in the air led to 50% reductions in cancer cell viability due to the cellular uptake of H_2_O_2_ and NO_2_^−^. Dahle et al. [57] reported that an increase in a plasma exposure time of up to 60 s in the air significantly reduced the number of Gram-negative bacterial cells due to OH radicals and excited N_2_ species. The result of this study indicated that the enhanced surface functionalization of 3Y-TZP could be obtained after exposure to air for 1–5 min by considering the amounts of OH(b), the polar components, and the nitrogen fixation involved in the plasma process. With longer exposure times, those values tended to be reduced, probably due to the increased ion density, which decreased the ion velocity and the collision frequency regime. However, in general, sustaining a stable and repeatable plasma reaction under atmospheric pressure is more difficult than at low pressures due to the high impedance of plasma gas in the air [58]. Vacuum plasma systems might have an advantage over atmosphere plasma systems in terms of manipulating the operational conditions (pressure, power, and gas flow). Nevertheless, atmospheric plasmas are more attractive because they can be easily handled in the air and have great potential in the production of active species. Therefore, in order to optimize the reliability of the atmospheric plasma treatment on the zirconia surface, the precise control of various process parameters, including the voltage, treatment time, and working gas, should be considered. The limitation of our study is that the percentage of N_2_ in the N_2_/Ar gas mixtures was set to 10%. Gas mixtures with different concentration ratios would affect electron collision cross-sections and nitrogen ion implantation processes. Our further study should deal with the effect of the N_2_/Ar gas mixture ratio on the plasma dynamics to selectively alter the zirconia surface properties.

## 4. Materials and Methods

### 4.1. Specimen Preparation and Plasma Surface Treatment

A total of 198 sintered 3Y-TZP specimens (KATANA ML, Kuraray Noritake Dental, Osaka, Japan) with dimensions of 10.0 mm × 10.0 mm × 1.0 mm were prepared. All of the specimens were polished using 600–1200 grit SiC abrasive papers and then cleaned in an ultrasonic bath of ethanol for 5 min. The 3Y-TZP specimens were randomly divided into two main groups according to the chamber gas pressure: vacuum plasma (V) and atmospheric plasma (A). Each group was subdivided into five subgroups according to the treatment time: 1, 5, 10, 15, and 20 min: V1, V5, V10, V15, and V20 for the vacuum plasmas; and A1, A5, A10, A15, and A20 for the atmospheric plasmas. The specimens of each experimental group were exposed to N_2_/Ar gas mixtures (10% N_2_ and 90% Ar) for varying treatment times under atmospheric or vacuum pressure. The control group was not subjected to the plasma treatment.

For the low-pressure groups, a planar ICP source (ICP system, Samvac Co., Paju, Republic of Korea) was powered with 150 W, −200 V_bias_ AC under 4 Pa of vacuum pressure at a 13.56 MHz radio frequency. For the air plasma generation, a DBD system (PR-ATO-001, ICD Co., Anseong, Republic of Korea) at low frequency (30 Hz) with alternating voltage (AC) was used. The distance between the plasma nozzle tip and the surface of the specimen was maintained at 10 mm. The schematic diagrams of the experimental setups are shown in Figure 7.

### 4.2. Contact Angle Measurements and Surface Free Energy Calculations

The surface free energy (*γ*) was obtained through contact angle measurements of 3 test liquids (water, glycerol, and diiodomethane) deposited on the zirconia surface using a contact angle analyzer (Phoenix 300 Touch, S.E.O., Suwon, Republic of Korea). According to the Lifshitz–van der Waals acid base method, *γ* can be divided into the additive Lifshitz–van der Waals (LW) and Lewis acid–base (AB) components [59]:(1)$$\gamma =\gamma^{LW}+\;\gamma^{AB}=\gamma^{LW}+\;2\sqrt{\gamma^{+}\gamma^{-}}$$
where $\gamma^{LW}$ includes all of the electrodynamical dispersion forces, and the acid–base components ($\gamma^{AB}$) break down into electron-donor ($\gamma^{-}$) and electron-acceptor ($\gamma^{+}$) parameters.

### 4.3. X-ray Photoelectron Spectroscopy (XPS) 

XPS analysis was performed to compare the changes in the surface chemistries of the atmospheric and vacuum plasma-treated zirconia with varying treatment times. The measurements were conducted for the core levels of C 1s, O 1s, N 1s, Y 3d, and Zr 3d regions using an XPS (K-alpha, Thermo Scientific Inc., Horsham, UK) equipped with a monochromatic Al Kα X-ray source (1486.6 eV) at 12 kV. The spectra were aligned to 284.6 eV of the C 1s peak as a reference. The compositional depth profiles of the plasma-treated zirconia surfaces were measured using Ar^+^ ion sputtering excited at an energy of 2 keV with a sputtering rate of 0.30 nm/s and a total sputtering time of 60 s to determine the permeation of nitrogen ions after the plasma treatments. 

### 4.4. Surface Topography 

The changes in the three-dimensional surface topographies after plasma irradiation were analyzed using CLSM (LEXT OLS3000, Olympus, Tokyo, Japan), with a measurement area of 256 × 192 μm^2^. The areal surface texture parameters (Sa, Sq, and Sv) were determined according to the ISO 25178 reference [60]. Ten measurements for each group were obtained.

### 4.5. SEM and FIB Analysis

The microstructures of the zirconia surfaces after the plasma treatments were characterized by SEM (JSM-7800F Prime, JEOL, Tokyo, Japan) at 2000×, 10,000×, and 40,000× magnifications. In combination with SEM, EDS spectra were obtained by using an EDS detector (X-max 150, Oxford Instruments NanoAnalysis, High Wycombe, UK) to determine the local chemical composition. To examine the subsurface structures, dual-beam cross-section analysis was performed with FIB/SEM imaging. The milling was carried out at a current of 300 pA using gallium ions accelerated at 30 kV. 

### 4.6. X-ray Diffraction (XRD) and Rietveld Analysis

The quantitative identification of the crystalline phases of each experimental group was determined by XRD (DMAX-2200PC, Rigaku, Tokyo, Japan) and a Rietveld refinement method with Cu*Kα* radiation at 40 KV and 30 mA. The XRD profiles were obtained at room temperature in the 2θ range of 10–100° at a step size of 0.02° and a counting time of 2 s per step. Structure refinements were performed by the Rietveld method with the Fullprof program [61]. The diffraction profiles were fitted with a pseudo-Voigt peak function and manually selected background points.

### 4.7. Measurement of Zeta Potentials

Zeta potential analyses were evaluated by an electrophoretic light-scattering technique in 10 mM NaCl (pH 5.6) using an electrokinetic analyzer (Mastersizer 3000, Malvern Panalytical Ltd., Malvern, UK). Five measurements were performed at 25 °C.

### 4.8. Statistical Analysis 

The statistical analysis of the data was performed using a two-way ANOVA to determine the effect of two independent variables, namely chamber pressure and plasma treatment time, on the contact angle, surface roughness, and zeta potential of 3Y-TZP. The analysis was carried out with a software suite (IBM SPSS Statistics, v25.0, IBM Corp., Chicago, IL, USA), and *p*-values below 0.05 were considered statistically significant.

## 5. Conclusions

In summary, both the vacuum and atmospheric plasma treatments changed the surface energy, chemical composition, and zeta potential of 3Y-TZP. Thus, the experimental results confirm that the plasma treatment of 3Y-TZP is an effective method for biomedical and clinical applications. The atmospheric plasma treatments increased zirconia’s electron donation ($\gamma^{-}$) capacity due to increased oxygen adsorption, while the vacuum plasma treatments decreased the $\gamma^{-}$ parameter with increasing treatment times. Higher concentrations of reactive oxygen species were found in atmospheric plasma groups compared to those in vacuum plasma groups due to the involvement of oxygen absorption from the air. The highest percentage of the OH(b) group was obtained after 5 min exposure to atmospheric plasmas. With longer exposure times, the vacuum plasmas induced physical or electrical damages, although these were confined to the outermost layer (≈top 10 nm). 

Both plasma systems increased the zeta potential of 3Y-TZP, showing positive values in a vacuum. In the atmosphere, the zeta potential rapidly increased after 1 min of exposure, and then a slight increase was observed with longer exposure times. In a vacuum plasma system, a much higher conversion and/or energy efficiency could be reached in only a short exposure time (<1 min). However, with a longer exposure, undesirable surface melting or degradation would be induced.

The enhanced surface functionalization of 3Y-TZP could be obtained after the exposure to air for 1–5 min by considering the amounts of OH(b), the polar components, and the nitrogen fixation involved in the plasma process. Atmospheric plasma treatments would be beneficial in terms of the adsorption of oxygen and nitrogen from ambient air and the generation of various active species on the zirconia surface, leading to the production of more active species than in a vacuum. 

## Figures and Tables

**Figure 1 ijms-24-07663-f001:**
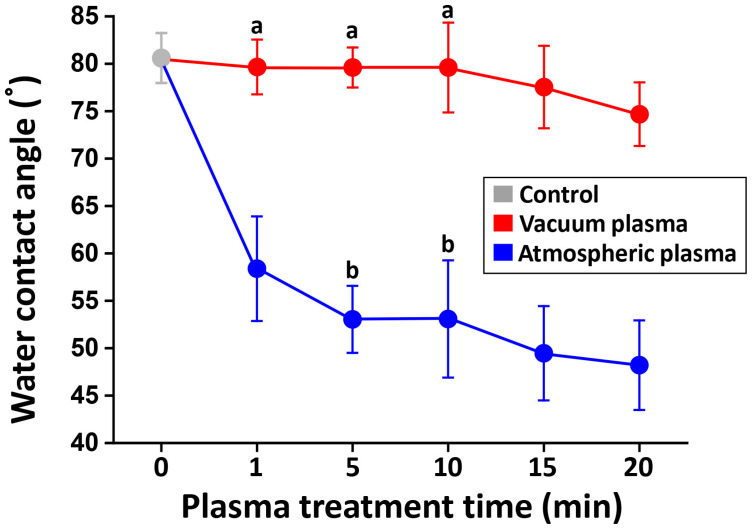
Changes in water contact angles as a function of plasma treatment time with atmospheric plasma and with vacuum plasma. The contact angle decreased with increasing treatment time for both plasma systems. The means within each plasma system that share identical letters are not significantly different from each other (*p* > 0.05).

**Figure 2 ijms-24-07663-f002:**
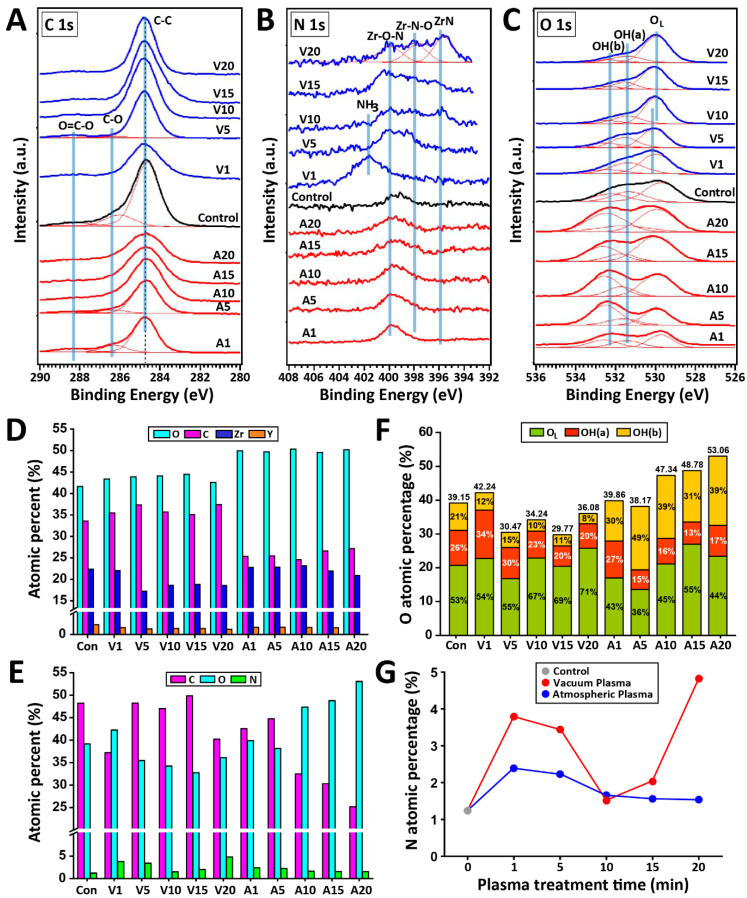
XPS spectra correspond to the C1s (**A**), N 1s (**B**), and O 1s (**C**) regions of all experimental groups. (**D**) Chemical composition obtained from SEM-EDS analysis. (**E**) Atomic percentages of C, N, and O on the zirconia surfaces were obtained from XPS spectra. (**F**) The relative ratios of lattice oxygen (O_L_), OH(a), and OH(b) in the O 1s core level XPS spectra. (**G**) N atomic percentage as a function of plasma treatment time obtained by XPS analysis.

**Figure 3 ijms-24-07663-f003:**
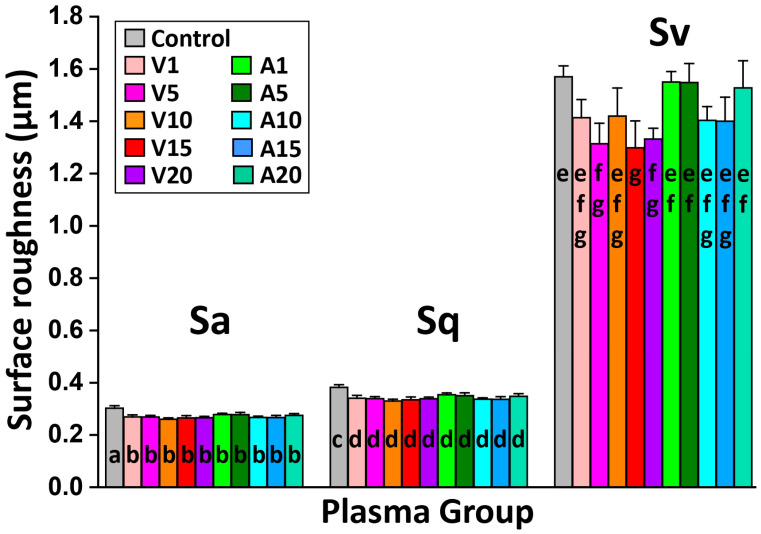
The surface texture parameters (Sa, Sq, and Sv) of all test groups. No significant differences in Sa, Sq, and Sv were observed among all plasma-treated groups. Means with identical letters are not significantly different from each other (*p* > 0.05).

**Figure 4 ijms-24-07663-f004:**
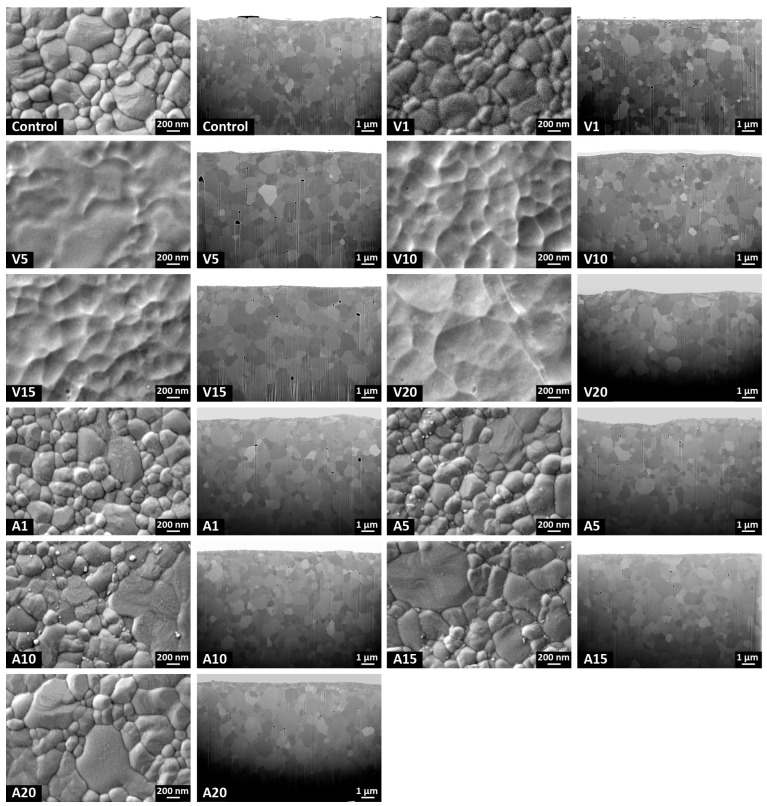
SEM images at 40,000× magnification (**left**) and FIB cross-sectional images of cross-sections at 6000× magnification (**right**) of each group. The vacuum plasma caused surface erosion due to electrical discharges, although it did not alter the subsurface microstructures.

**Figure 5 ijms-24-07663-f005:**
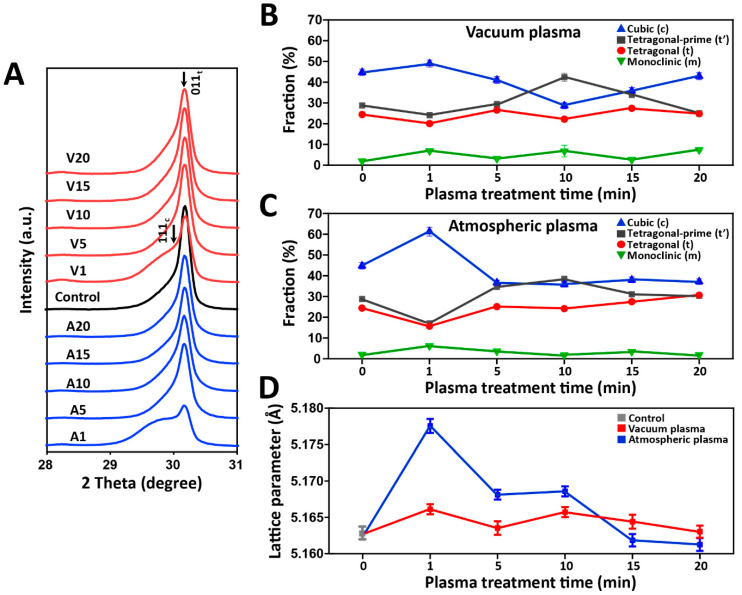
(**A**) X-ray diffraction patterns of all experimental groups; Rietveld quantitative analyses as a function of the plasma exposure time for vacuum plasma groups (**B**) and atmospheric plasma groups (**C**); (**D**) The unit cell parameter (Å) of cubic phase as a function of the plasma exposure time.

**Figure 6 ijms-24-07663-f006:**
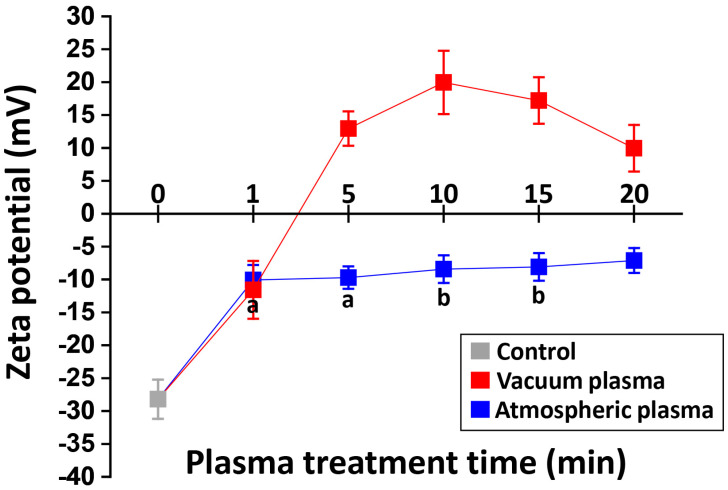
Changes in zeta potential as a function of the plasma treatment time with atmospheric plasma and with vacuum plasma. The zeta potentials of control, atmospheric plasma groups, and V1 are negative, while those of the vacuum plasma groups except V1 are positive.

**Figure 7 ijms-24-07663-f007:**
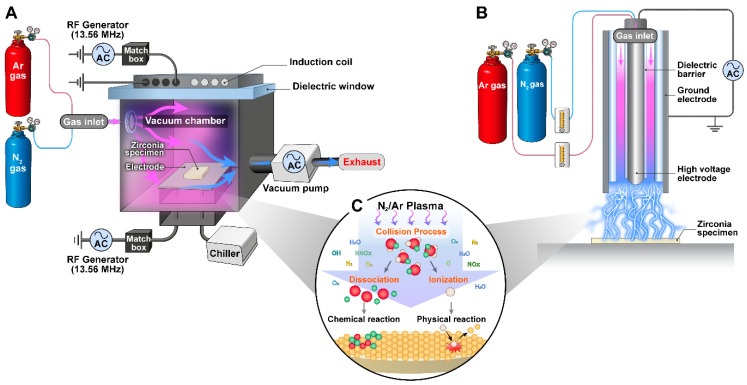
Schematic diagrams of experimental setups for plasma surface modifications: (**A**) ICP vacuum plasma system, (**B**) DBD atmospheric plasma system, and (**C**) mechanism of the plasma surface modification of 3Y-TZP.

**Table 1 ijms-24-07663-t001:** Surface energy components of all experimental groups are based on probe liquids. Values are in mJ/m^2^.

Material	*γ*	$\gamma {\;}^{LW}$	$\gamma^{AB}$	$\gamma^{+}$	$\gamma^{-}$
**Liquid**					
DI water ^a^	72.80	21.80	51.00	25.50	25.50
Glycerol ^a^	63.40	34.00	29.40	3.92	57.00
Diiodomethane ^a^	50.80	50.80	0.00	0.00	0.00
**Group**					
Control	17.51	41.40	−23.89	5.98	23.84
V1	33.98	39.77	−5.79	0.28	29.99
V5	34.62	35.56	−0.94	0.02	9.91
V10	34.63	36.66	−2.03	0.11	9.71
V15	39.17	37.44	1.74	0.10	7.83
V20	30.64	35.61	−4.97	0.35	17.56
A1	30.74	41.36	−10.61	0.69	40.82
A5	34.97	39.65	−4.67	0.14	38.37
A10	36.62	37.48	−0.86	0.01	33.02
A15	36.57	39.65	−3.08	0.06	42.17
A20	31.94	40.53	−8.59	0.40	45.68

Superscripts *LW* and *AB* account for Lifshitz–van der Waals and Lewis acid–base interactions (electron acceptor, $\gamma^{+}$/donor, $\gamma^{-}$), respectively. ^a^ Values are proposed by van Oss [22].

## Data Availability

The data presented in this study are available upon request from the corresponding author.

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
