# Peer review of "Plasma Surface Modification of 3Y-TZP at Low and Atmospheric Pressures with Different Treatment Times"

_ijms, 2023, doi:10.3390/ijms24087663_

Round 1

Reviewer 1 Report

Dear Miss Yi Cai, dear authors of the manuscript,

After careful review of the manuscript submitted by Sung Un Kan and Chul-Ho Kim I’d like to provide following report:

The authors present a systematic study on plasma processes applying different pressure regimes and different treatment times on yttria-stabilized zirconia samples in order to modify the samples’ surfaces. Different surface analysis methods were applied to investigate the impact of the plasma treatments on the chemical and physical surface characteristics. The manuscript is written well, results are presented reasonable and conclusions derived from results are plausible and understandable.

The introduction comprehensively covers plasma science and biomedical applications of the treated samples. In the results passage different characterization methods are well presented, magnified views on certain important aspects, e.g. Fig. 5A, are chosen well. The discussion covers most important results holistically. Particularly noteworthy is the discussion of apparatus-related influences on the gathered results, as different devices/setups have been used for plasma treatments at different pressures.

As such, I’d recommend to accept the manuscript by Sung Un Kan and Chul-Ho Kim for publication with some minor corrections:

·        Please place the conclusion part directly after the discussion part. I’d also encourage the authors to add a small passage in the conclusion part whether the gained results are beneficial or detrimental for biomedical applications of the plasma treated material.

·        XRD-Investigations: It would be helpful to follow the XRD discussion if images of the different unit cells of the different crystal structures of YSZ along with the diffraction patterns were shown in the SI. Please add unit cell images to the SI.

·        Figure 5, caption: Association of figure 5 D is missing. Please add (D) to the caption

·        Analysis of variance (ANOVA) should be given in plain text when mentioned for the first time

·        Please use SI units for pressure, i.e. Pa instead of mTorr.

·        Please use uniform formatting when listing all references, i.e.  Surf. Coat. Technol.  Instead of Surf Coat Technol and please add an R at the end of journal name in reference 58, i.e.  Matter instead of Matte

Best regards

Reviewer 2 Report

1. In Fig.1, what is the meaning of label a and b?

2. The N atomic percentage in vacuum plasma treated samples increased after 10 to 20 minutes’ treatment. Why?

3. The manuscript’s structure should be re-constructed. Part 4 of materials and methods should be in front of the results.

4. To better understanding of the nitriding effects, the EDS along the cross-section of the plasma-treated samples should be characterized.

5. The schematic diagram should be provided to better understanding of the mechanism of the plasma surface modification of 3Y-TZP.   

Round 2

Reviewer 2 Report

1. It is essential to characterize the element dispersing across the cross-section of the modified samples. The permeation of nitrogen ion from the top surface exits difference after the activation of nitrogen atoms. It should be characterized and discussed clearly.

2. Ref 27 is not the authors work. It is only a reference. In results part, it can be used as comparison and can not be used as the authors research results of the elements on the modified samples surfaces. 

Author Response

Dear Reviewer and Editor,

We, the authors, highly appreciate the detailed valuable comments on this manuscript.

The revision was listed below the comments and recommendations one by one.

We also highlighted the revised parts in red color for your convenience in the revised paper.

================================================================

Response to Reviewer 2 Comments

Point 1: . It is essential to characterize the element dispersing across the cross-section of the modified samples. The permeation of nitrogen ion from the top surface exits difference after the activation of nitrogen atoms. It should be characterized and discussed clearly.

Response 1:

I am sorry that we did not investigate the dispersion of chemical elements across the cross-section of zirconia surfaces. We should perform the nitrogen diffusion measurements for our further study:

~The SEM-EDS analysis limits the nitrogen detection due to the low efficiency for low-Z elements [27].~

~Another limitation is that we did not investigate the dispersion of chemical elements across the cross-section of zirconia surfaces to determine the permeation of nitrogen ion after the plasma treatments. Since the diffusion of nitrogen gas might depend on the presence of polar and hydrophilic functional groups, our further study should perform the nitrogen diffusion measurements with the plasma treatments.

Point 2: . Ref 27 is not the authors’ work. It is only a reference. In results part, it can be used as comparison and can not be used as the authors’ research results of the elements on the modified samples’ surfaces.  

Response 2:

We removed ‘Ref 27’ in the revised manuscript. Thank you.

~The chemical composition (of a region about 1 μm in depth) obtained from~

Round 3

Reviewer 2 Report

The permeation of nitrogen ion from the top surface to the substrate reflects the surface modification degree. It is essential and crucial to characterize the element dispersing across the cross-section of the modified samples. It should be characterized and discussed clearly. The line scanning of elements across the cross-section of the modified samples could and should be supplied. 

Round 4

Reviewer 2 Report

Accept.